# Low-Temperature Processed TiO_x_/Zn_1−x_Cd_x_S Nanocomposite for Efficient MAPbI_x_Cl_1−x_ Perovskite and PCDTBT:PC_70_BM Polymer Solar Cells

**DOI:** 10.3390/polym11060980

**Published:** 2019-06-03

**Authors:** Binh Duong, Khathawut Lohawet, Tanyakorn Muangnapoh, Hideki Nakajima, Narong Chanlek, Anirudh Sharma, David A. Lewis, Pisist Kumnorkaew

**Affiliations:** 1National Nanotechnology Center, 111 Thailand Science Park, Paholyothin Road, Klong 1, Klong Luang, Pathumthani 12120, Thailand; bduong07@gmail.com (B.D.); khathawut.loh@nanotec.or.th (K.L.); tanyakorn.mua@nanotec.or.th (T.M.); 2Synchrotron Light Research Institute, 111 University Avenue, Muang District, Nakhon Ratchasima 30000, Thailand; hideki@slri.or.th (H.N.); narong@slri.or.th (N.C.); 3University of Bordeaux, Laboratoire de Chimie des Polymères Organiques (LCPO), UMR 5629, B8 Allée Geoffroy Saint Hilaire, 33615 Pessac Cedex, France; anirudh.sharma@enscbp.fr; 4Flinders Institute for Nanoscale Science and Technology, Flinders University, Adelaide, SA 5042, Australia; david.lewis@flinders.edu.au

**Keywords:** electron transport layer, TiO_x_, ZnCdS, solar cells, perovskite, polymer, scanning Kelvin probe microscopy

## Abstract

The majority of high-performance perovskite and polymer solar cells consist of a TiO_2_ electron transport layer (ETL) processed at a high temperature (>450 °C). Here, we demonstrate that low-temperature (80 °C) ETL thin film of TiO_x_:Zn_1−x_Cd_x_S can be used as an effective ETL and its band energy can be tuned by varying the TiO_x_:Zn_1−x_Cd_x_S ratio. At the optimal ratio of 50:50 (vol%), the MAPbI_x_Cl_1−x_ perovskite and PCBTBT:PC_70_BM polymer solar cells achieved 9.79% and 4.95%, respectively. Morphological and optoelectronic analyses showed that tailoring band edges and homogeneous distribution of the local surface charges could improve the solar cells efficiency by more than 2%. We proposed a plausible mechanism to rationalize the variation in morphology and band energy of the ETL.

## 1. Introduction

The ever-growing energy demand relies mainly on the combustion of fossil fuel, which continues to cause serious resource depletion and environmental pollution. Solar energy is a proven renewable energy that is environmentally friendly and free from regional restriction. Inorganic solar cells (i.e., Si-based, gallium arsenide, copper indium gallium selenide, cadmium telluride, etc.) offer high efficiency (more than 25%) and stability, however, the technology development is hindered by a sophisticated manufacturing process in addition to the handling of toxic materials [1]. While dye-sensitized, polymer and perovskite solar cells are the emerging photovoltaic devices that are both lightweight and low cost, they however have their own set of problems that inhibit their large-scale production as was highlighted in recent review papers [1,2,3]. To realize their practical applications, device stability is one of the critical factors that needs to be understood and overcome. Different approaches have been proposed for a more stable polymer [4,5,6], dye-sensitized and perovskite solar cells [7,8,9,10,11].

Interface engineering can be considered as the most crucial factor in order to achieve high performance and stable solar cells, because it shapes the pathways of photogenerated charge carriers from an active layer to metal electrodes. Both polymer solar cells and perovskite solar cells can be easily fabricated in n-i-p configuration where electrons are collected at the bottom electrode, or p-i-n configuration where electrons are collected at the top electrode, via simple solution processes. In terms of device performances, p-i-n devices often exhibit more reliable performances and better environmental stability. However, their efficiencies are slightly lower than that of n-i-p structures due to the barrier in Fermi levels at the interface between the electron transport layer and the metal electrode [12]. Significant efforts have been devoted to developing interfacial materials for better band alignment, defect passivation and charge separation. While ZnO [13,14], SnO_2_ [15], Al_2_O_3_ [16], MoO_x_ [17], core-shell nanoparticles/structures [18], and organic compounds [3,19,20] have all been reported as materials for electron transport layer (ETL), TiO_2_ remains the most widely used and essential ETL material for both polymer solar cells and perovskite solar cells owing to the ease of tuning its optical and electrical band gaps. Comprehensive reviews on ETL materials, properties and interface engineering have been reported in recent papers [12,21,22]. Different concepts of tailoring TiO_2_ have been proposed and showed promising results including metal-doped TiO_2_ [23], amino acid-modified TiO_2_ [24], plasmon-mediated TiO_2_ [25], or functionalized organic/polymer on TiO_2_ [26,27].

Instead of doping TiO_2_, synthesizing ETL composite of two metal oxides with appropriate band edges is another approach in facilitating electron transport, offering better efficiency and minimizing recombination rate. ETL composites of SrTiO_3_/TiO_2_ [28], BaTiO_3_/TiO_2_ [29], and In_2_O_3_/TiO_2_ [30] have been shown to improve the overall device efficiency. Nevertheless, it is critical to further optimize the suitable mixing ratio (molar ratio, volume ratio) in order to achieve high power conversion efficiency. Controlling film thickness and morphology are the main challenges in using nanocomposites as ETL. It is also unclear how work function would vary at nanometer level when two semiconducting metal oxides are mixed together and the influence of lateral inhomogeneity of work function on the device performance. 

In addition, the formation of crystalline TiO_2_ requires high temperature annealing (500 °C) which is not suitable for fabricating devices on flexible substrates, as well as for many other practical applications. Several groups have addressed this issue by developing low-temperature processes. Kim et al. found that the devices made of ultra-thin amorphous TiO_x_ by atomic layer deposition are more stable than crystalline TiO_2_ [31,32]. Recently, Deng et al. reported that high efficiency perovskite solar cells could be achieved using TiO_x_ processed at room-temperature [33]. Another study by Wang et al. demonstrated that efficient perovskite devices made of ETL nanocomposites of graphene/TiO_2_ could be processed at 150 °C [34]. 

Here, we show that low-temperature (80 °C) processed TiO_x_:Zn_1−x_Cd_x_S (T:Z) can be used as an effective ETL in both p-i-n perovskite and polymer solar cells. Zn_1−x_Cd_x_S was chosen due to the ease of synthesis and highly tunable bandgap by varying Zn/Cd molar ratios [35]. We examined performances of the p-i-n devices using different loadings of Zn_1−x_Cd_x_S in TiO_x_ and carried out electrochemical impedance spectroscopy, ultraviolet photoelectron spectroscopy (UPS) and scanning Kelvin probe microscopy (SKPM) measurements. The local morphology with corresponding work function obtained from SKPM can shed more light into the fundamental understanding of electron transport layer in solar cells.

## 2. Materials and Methods

### 2.1. Chemical Preparation

#### 2.1.1. Zn_1−x_Cd_x_S Nanoparticles Synthesis

During the process, 0.55 g of zinc acetate (99.99%, Ajax Finechem, New South Wales, Australia) and 0.33 g of cadmium acetate (99.99% Asia Pacific Specialty Chemicals Limited, New South Wales, Australia) were dissolved in 25 mL of DI water and stirred for 15 min. Another solution of 0.65 g thioacetamide (99.0%, Sigma Aldrich, MO, USA) and 0.6 g of polyvinylpyrrolidone (99.0%, Sigma Aldrich, MO, USA) in 25 mL DI water were prepared. Both solutions were mixed and stirred at room temperature for 30 min. The mixture was heated with 700 W microwave (Sharp Cooperation, Osaka, Japan) radiation for 3 min. Zn_1−x_Cd_x_S nanopowder was filtered and rinsed with 2-butanol [36]. The nanopowder was dried at 250 °C for 2 h and calcined at 500 °C for 1 h. Suspension of 0.4 mg/mL Zn_1−x_Cd_x_S in 2-butanol was used in all experiments.

#### 2.1.2. TiO_x_ Synthesis

During the synthesis, 2.7 mL titanium (IV) butoxide (97%, Sigma Aldrich, MO, USA), 0.57 mL ethyl acetoacetate (99%, Sigma Aldrich), 57 μL 2,4-pentanedione (99%, Sigma Aldrich, MO, USA) and 57 μL 2,3-butanedione (97%, Sigma Aldrich, MO, USA) were mixed and stirred at room temperature overnight [37].

#### 2.1.3. Polymer (PCDTBT:PC_70_BM) Ink Formulation

For PCDTBT:PC_70_BM ink formation, 8 mg of PCDTBT (Ossila Ltd., Sheffield, UK) and 32 mg of PC_70_BM (Ossila Ltd., Sheffield, UK) were dissolved in 1 mL anhydrous dichlorobenzene (Sigma Aldrich, MO, USA). This *PCDTBT:PC_70_BM* is used for polymer solar cells. For perovskite solar cells, PC_70_BM solution was prepared by dissolving 50 mg of PC_70_BM into 1 mL anhydrous dichlorobenzene (Sigma Aldrich, MO, USA). Both solutions were then stirred at 60 °C for 1 h [17].

#### 2.1.4. Perovskite (MAPbI_x_Cl_3-x_) Ink Formulation

For MAPbI_x_Cl_3-x_ ink formation, 2.4 g of MAI (Ossila Ltd., Sheffield, UK) and 1.4 g PbCl_2_ (Sigma Aldrich, MO, USA) were mixed and stirred in 5 mL anhydrous dimethylformamide (Sigma Aldrich, MO, USA) at 70 °C for 2 h [38]. 

### 2.2. Device Fabrication

Polymer and perovskite solar cells were fabricated on pre-patterned ITO glass (Luminescence Technology Corporation, Taipei, Taiwan, 25 × 25 mm^2^, 5 Ω/sq). Prior to the fabrication, the substrates were cleaned by ultrasonication under detergent, DI water, acetone, and isopropyl alcohol sequentially. The substrates were further exposed under 395 UV light for 20 min.

Convective deposition [39] or spin coating method was used for depositing of hole transporting, photoactive, and electron transporting layers. A cleaned glass microscope slide (75 × 25 × 1 mm^3^, Fisher PA, Charlotte, NC, USA) was used as a deposition blade and placed at 45° with respect to the patterned ITO substrate. PEDOT:PSS was used as the hole transporting layer for both solar cells. 20 μL of PEDOT:PSS (100 nm) were coated twice by convective deposition at the speed of 3000 μm/s. The coated substrate was annealed on a hot plate at 120 °C for 30 min in the ambient environment (%RH 50-60). The polymer solar cell fabrication was continued under ambient conditions while the PEDOT:PSS coated substrates were transferred to a glove box for perovskite solar cell fabrication. 

For polymer solar cell with ITO/PEDOT:PSS/PCDTBT:PC_70_BM/TiO_x_/Al structure, 20 μL PCDTBT:PC_70_BM (80 nm) were convectively deposited at the speed of 750 μm/s and left in air for 3 min followed by the deposition of 20 μL TiO_x_ solution with various Zn_1−x_Cd_x_S suspension ratio. The coated films (20 nm) were annealed at 80 °C. 

For perovskite solar cell with ITO/PEDOT:PSS/CH_3_NH_3_PbI_x_Cl_3-x_/PC_70_BM/TiO_x_/Al structure, 40 μL perovskite ink were deposited with spin coating technique with 2250 rpm for 30 s. The substrate and ink were kept at 70 °C before the deposition. The perovskite (300 nm) coated substrate was annealed at 100 °C on a hot plate for 90 min. Following this, 35 μL of 50mg/mL PC_70_BM solution were spin casted onto the perovskite film at 1500 rpm for 30 s. Then, 200 μL TiO_x_ solution with various Zn_1−x_Cd_x_S suspension ratio were spin casted (20 nm) with 2500 rpm for 30 s. The multilayer films were then annealed at 80 °C for 30 min in a glove box. 

Aluminum metal electrode (80 nm) was deposited on top of the TiO_x_ layer to complete the polymer and perovskite solar cell fabrication. Four independent cells with 0.1 cm^2^ active area per cell were fabricated on the patterned ITO substrate. 

### 2.3. Characterization

Surface morphologies were characterized by atomic force microscope (NanoWizard 3 NanoScience, JPK Instruments, Berlin, Germany).). Bruker OSCM-PT-R3 AFM probes (Bruker Nano, CA, USA) were used. The solar cell performance was measured using 0.1 cm^2^ mask under 100 mW/cm^2^ and 1.5 AM filter (Abet Technologies, Milford, CT, USA). The light intensity was calibrated by 4 cm^2^ solar cell reference (VLSI Standards, Mountain View, CA, USA). I-V measurements were conducted using source measurement unit (PXI-4130, National Instrument, Austin, TX, USA with 10 mV scan step from 1 V to −1 V. Incident photon-to-current efficiency IPCE was measured using QEPVSI-b (Newport Corporation, Irvine, CA, USA). Impedance spectroscopy and Mott-Shockey (M-S) measurements were carried out using CHI 660E electrochemical work station (CH Instruments, Bee Cave, TX, USA). For M-S measurement, a small AC voltage of 5 mV was applied under constant illumination to measure device impedance as a function of frequency (100 Hz to 1 MHz). UPS was carried out at the Synchrotron Light Research Institute.

## 3. Results and Discussion

To determine the best loading ratio of Zn_1−x_Cd_x_S in TiO_x_ in both polymer and perovskite solar cells, we fabricated p-i-n cell structures using various volumetric mixing ratios of TiO_x_:Zn_1−x_Cd_x_S dispersions as 100:0 (T100), 75:25 (T75:Z25), 50:50 (T50:Z50), 25:75 (T25:Z75), 0:100 (Z100). Figure 1a shows SEM cross-sectional image of the device structures. The perovskite solar cells made of T100, T75:Z25, T50:Z50, T25:Z75 and Z100 achieved the efficiencies of 7.74%, 9.47%, 9.79%, 8.07% and 6.74%, respectively. Polymer solar cells share similar trends with perovskite devices, namely 4.22% (T100), 4.40% (T75:Z25), 4.95% (T50:Z50), 3.63% (T25:Z75) and 2.93% (Z100). Incorporating Zn_1−x_Cd_x_S shows an increase in current density from 15 mA/cm^2^ to 18 mA/cm^2^ for perovskite solar cells (Figure 1b) and from 8 mA/cm^2^ to 11 mA/cm^2^ for polymer solar cells (Table 1) and Appendix A indicating better carrier extraction from the active layer after ETL modification. Further details on perovskite solar cell performances are shown in Table 2. Negligible hysteresis for this p-i-n perovskite solar cell was observed after forward-reverse scan study as shown in Appendix A. The results agree well with previous studies [40,41,42].

EIS and Mott-Schotkky measurements were carried out to understand the recombination losses and charge transport properties upon mixing Zn_1−x_Cd_x_S into TiO_x_. Figure 1c shows Nyquist plots for perovskite solar cells and polymer solar cells at different loading of Zn_1−x_Cd_x_S. The high frequency intercept of the Nyquist plot attributed to the series resistance and the low-frequency intercept corresponds to the recombination resistance (R_rec_). As R_rec_ gets larger, lower leakage current and recombination occur at the surface. It was clearly observed that R_rec_ increased for devices containing Zn_1−x_Cd_x_S, which had lower recombination losses. In addition, the lifetime of free carriers (*τ*) could be calculated from the maximum frequency (*f_m_*) obtained from Bode plot as shown in Figure 1d by
(1)$$\tau =\frac{1}{2\pi f_{m}}$$

The carrier lifetime increased from 46 μs to 68 μs and to 82 μs as Zn_1−x_Cd_x_S content increased from 0 to 100%. It is also well known that the interfacial charge density is inversely proportional to the slope of the Mott-Schottky plots (Figure 1e) as follows:(2)$$\frac{1}{C^{2}}=\frac{2}{e\varepsilon_{o}\varepsilon_{r}NA^{2}}(V-V_{fb}-\frac{k_{B}T}{e})$$
where *C* is capacitance, *ε*_o_ is vacuum permittivity, *ε*_r_ is dielectric constant of material, *N* is charge carrier density, *A* is electrode area, *V* is applied potential, *V_fb_* is flat band potential, *k_B_* is Boltzmann constant, *T* is absolute room temperature and *e* is elementary charge of electron. As the charge carrier density is smaller than 5 × 10^20^ cm^−3^ [43], the difference in flat band potential is negligible. Assuming an equivalent dielectric constant for all devices under illumination, the slopes of T100, T75:Z25, T50:Z50, T25:Z75 and Z100 are −2.2 × 10^10^, −2.3 × 10^10^, −2.4 × 10^10^, −1.8 × 10^10^, −1.0 × 10^10^ respectively, indicating the highest interfacial charge density and resulting in the highest current density.

To better understand the nature of the starting materials, we performed detailed characterizations of TiO_x_, Zn_1−x_Cd_x_S and their nanocomposites. SEM images and EDS spectra (Appendix A) to confirm that the prepared nanocomposites form uniform, smooth thin films and addition of Zn_1−x_Cd_x_S in TiO_x_ does not effect on the morphology of the films. TEM images (Appendix A) reveal that the TiO_x_ is in amorphous form with small amount of anatase crystals of TiO_2_ while Zn_1−x_Cd_x_S is fully crystalline with crystallite size of about 5 nm. UV-Vis spectra and Tauc plots are shown in Appendix A. The band gaps calculated from Tauc plot of T100 and Z100 thin films are 3.7 and 2.5 eV, respectively, which are in good agreement with the band gap of TiO_x_ and Zn_1−x_Cd_x_S reported in the literatures [33,35]. The band gap values obtained from highest slopes in the Tauc plot of T75:Z25, T50:Z50, T25:Z75 films share the same value of 3.47 eV, which is lower than that of the pure T100 film. The nanocomposite ETL films also exhibit minor slopes, indicating indirect transitions or existence of gap states [44]. The indirect band gap is shifted from 3.0 eV for T75:Z25 to 3.1 eV for T50:Z50 to 3.25 eV for T25:Z75.

Ultraviolet photoelectron spectroscopy (UPS) was used to characterize the work function and valence band energy of TiO_x_:Zn_1−x_Cd_x_S electron transport layer. UPS spectra of 10 nm T:Z thin films coated on silicon substrates are shown in Figure 2a. The effective work functions of the films were obtained by subtracting the source energy (40.8 eV) by the energy difference (*ΔKE*) between the low kinetic energy edge (*LKE*) and the spectrometer/sample Fermi level:(3)ϕ=40.8−ΔKE

The valence band energy (*E_VB_*) was estimated by subtracting the source energy (40.8 eV) by the energy difference (*ΔKE’*) between the LKE and the high kinetic energy (*HKE*) edge for photoemission as below:(4)$$\phi =40.8-\Delta KE^{\prime }$$

The work functions of T100, T75:Z25, T50:Z50, T25:Z75 and Z100 thin films with respect to the vacuum level are 4.22, 4.00, 3.70, 4.08 and 3.60 eV, respectively (Appendix A). The valence band energy of T100, T75:Z25, T50:Z50, T25:Z75 and Z100 thin films with respect to the vacuum level are −7.42, −7.20, −6.90, −7.28 and −7.80 eV, respectively. Figure 2b shows a proposed general band energy diagram of TiO_x_:Zn_1−x_Cd_x_S nanocomposite electron transport layer. The band energy levels of ITO, PEDOT, perovskite/PCDTBT, PC_70_BM and Al were taken from other studies [12,32].

Scanning Kelvin probe microscopy (SKPM) is an effective technique to monitor simultaneously morphology/topography and contact potential difference (V_CPD_) between the atomic force microscopy tip and sample surface. When the AFM tip and sample is far apart and not connected electrically, their Fermi levels are different. As the AFM tip approaches the sample surface, an electrical force is induced between the tip and the sample surface, brought their Fermi levels to line-up and formed contact potential difference (CPD). To measure this CPD, an external bias is applied at the contact area until the surface potential difference between the tip and the sample becomes zero. The magnitude of the applied bias equals to the work function difference between the tip and the sample, which means the work function of the sample can be calculated once the work function of the tip is known. The measured CPD is defined as the difference in work function between a tip and a sample as follows:(5)$$V_{CPD}=\frac{\emptyset_{tip}-\emptyset_{sample}}{e^{-}}$$
where V_CPD_ is the CPD between the tip and the sample, φ_tip_ and φ_sample_ are the work functions of the tip and the sample, respectively and e^-^ is the elementary charge. To obtain the work function of the AFM tip, freshly cleaved highly oriented pyrolytic graphite (HOPG, φ = 4.6 eV) and thermally evaporated gold thin film (Au, φ = 5.1 eV) were used. For all measurements, the substrate was grounded. Topographical and potential maps of the pure TiO_x_, Zn_1−x_Cd_x_S and mixed TiO_x_:Zn_1−x_Cd_x_S thin films are shown in Figure 3. Three-dimensional maps combining topographic and V_CPD_ data clearly show that the T50:Z50 film possesses the least variation in surface potential compared to other ETL films. Potential histograms of all samples are shown in Figure 4a. The work function of T50:Z50 is 3.75 ± 0.28 eV compared to 3.47 ± 0.44 eV of T100 and 3.77 ± 0.29V eV of Z100, which are comparable to UPS data. FWHM value of T50Z50 has minimum value, indicating the least variation in work function at nanoscale level. It is expected that the work function values of ETL composite films obtained from SKPM technique are slightly lower than those obtained from UPS method due to water adsorption and hydrocarbon contamination (Appendix A). Although the work function value of T100 film estimated from SKPM method is about 0.4 eV lower than that of UPS value, it is in good agreement with plasma enhanced atomic layer deposited TiO_x_ films reported in Kim’s study [32]. Figure 4b–f shows profiles at three different locations from SKPM image of T25:Z75 film, which indicates that the surface potential depends on the film roughness. As the tip scanned over a particle, the height abruptly increased about 5 nm, and the local V_CPD_ decreased about 200 mV, leading to an increase at the local work function. The distribution of charges at the surface and potential step can be attributed to crystallographic orientation and atomic relaxation [45]. It is possible that electronegative character of Zn_1−x_Cd_x_S results in a partial electron transfer with TiO_x_ that leads to an increase in local work function [45]. When the height decreased about 3 nm, the local V_CPD_ increased about 100 mV. Within 1–2 nm variation in height, the local charge distribution could be fluctuated about 50 mV. The overall results indicate that an effective ETL composites of mixed metal oxides should have root mean square roughness of about 0.5 nm and scanning probe microscopy can be used as an effective tool to predict the homogeneity of local charges.

We suggest the following mechanism to rationalize for morphological and electronic variation of TiO_x_:Zn_1−x_Cd_x_S nanocomposite films. In pure TiO_x_ film, nanocrystals can be formed from Ti-O monomers via hydrolysis and condensation reactions which are strongly dependent on reaction temperature. Solvent evaporates, supersaturates the polymeric precursor solution and leads to the formation of a large number of Ti-O species. At low temperature (80 °C), the nuclei impinge on each other and impede further grain growth via diffusion, resulting in a relatively smooth film of small grains (RMS: 0.7 nm), but the mixture of amorphous and crystalline phases (dominantly amorphous phase) leads to broad distribution of work function (3.47 ± 0.44 eV). Pure Zn_1−x_Cd_x_S film follows similar mechanism as TiO_x_ film, however, there is no polymeric precursor. The pure Zn_1−x_Cd_x_S film is in smooth, crystalline phase (RMS: 0.9 nm), resulting in narrow distribution of work function (3.77 ± 0.29V). When preexisting Zn_1−x_Cd_x_S seeds are present in the initial stage of the synthesis, the Ti-O species are immobilized onto the seed crystals to overcome the activation energy barrier for nucleation, leading to the formation of bigger grains compared to that of pure films and creating an ordered electronic network of TiO_x_ surrounding the grains. For T75Z25 film, the density of Zn_1−x_Cd_x_S in the composite film is too low, resulting in rougher film due to scattering of the large crystalline grains. For T25Z75 film, the density of Zn_1−x_Cd_x_S in the composite film is too high, the grains are interconnected by the polymer chains similar to the mechanism proposed by Yang’s group [11], however, the volume of TiO_x_ is not enough to bridge the gaps between grains, resulting in a rough surface. For T50Z50 film, the density of Zn_1−x_Cd_x_S is optimal in the TiO_x_ matrix, the distribution of seed crystals and its proximity interaction with surrounding polymer chains is the most effective, resulting in the smoothest film among all. As the morphology and internal structures of ETL films are tuned by addition of Zn_1−x_Cd_x_S, the band energy of ETL is simultaneously modified due to presence of Zn_1−x_Cd_x_S energy states and surrounding TiO_x_ electronic network, which are supported by UV-Vis, UPS and SKPM results. Ultimately, appropriate addition of Zn_1−x_Cd_x_S can be used to maximize the optoelectronic output and stability of ETL film and the corresponding devices.

## 4. Conclusions

This study has demonstrated that efficient perovskite and polymer solar cells were successfully fabricated using low-temperature processed TiO_x_:Zn_1−x_Cd_x_S thin film as an electron transport layer. The ETL band energy can be simply tuned by varying TiO_x_:Zn_1−x_Cd_x_S ratio. The addition of Zn_1−x_Cd_x_S in TiO_x_ lowers band gap, alters valence band positions and work function levels, reduces recombination at interfaces and increases charge carrier lifetime. The impact of local surface potential homogeneity of TiO_x_:Zn_1−x_Cd_x_S nanocomposite electron transport layer on the performance of p-i-n perovskite and polymer solar cells has been revealed. Together with tailoring band edges, a homogeneous distribution of the local surface charges has been shown to improve solar cells’ efficiency by more than 2%.

## Figures and Tables

**Figure 1 polymers-11-00980-f001:**
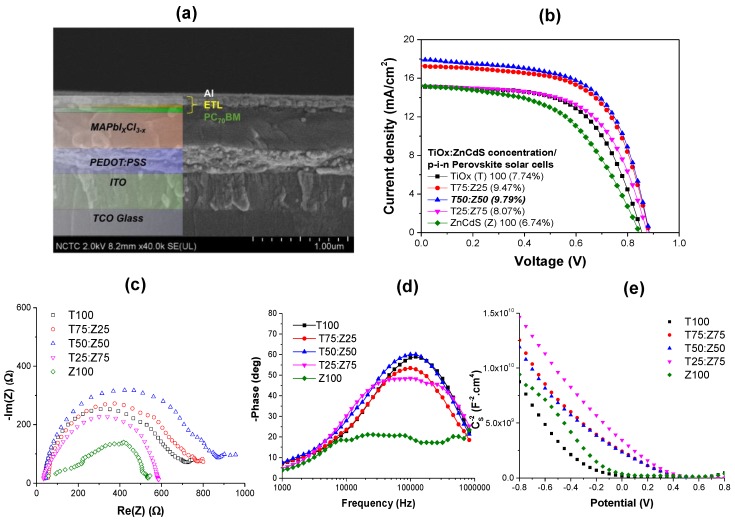
(**a**) SEM cross-sectional image, (**b**) J-V characteristics of perovskite solar cells, (**c**) Nyquist plot, (**d**) Bode plot and (**e**) Mott-Schottky plot of perovskite solar cells fabricated from different ratio of TiO_x_:Zn_1−x_Cd_x_S.

**Figure 2 polymers-11-00980-f002:**
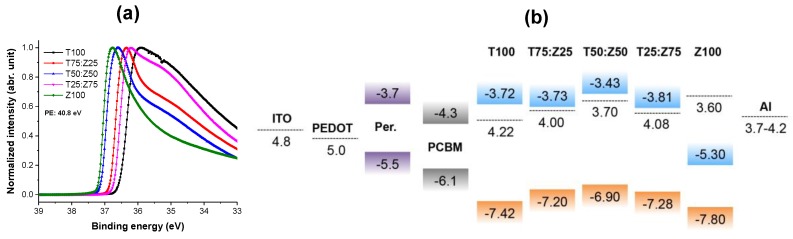
(**a**) UPS spectra and (**b**) proposed energy band diagram of TiO_x_:Zn_1−x_Cd_x_S nanocomposite films. Energy levels of ITO, PEDOT:PSS, perovskite, PC_70_BM and Al were taken from other studies [3,32]. Photon energy (PE) is 40.8 eV.

**Figure 3 polymers-11-00980-f003:**
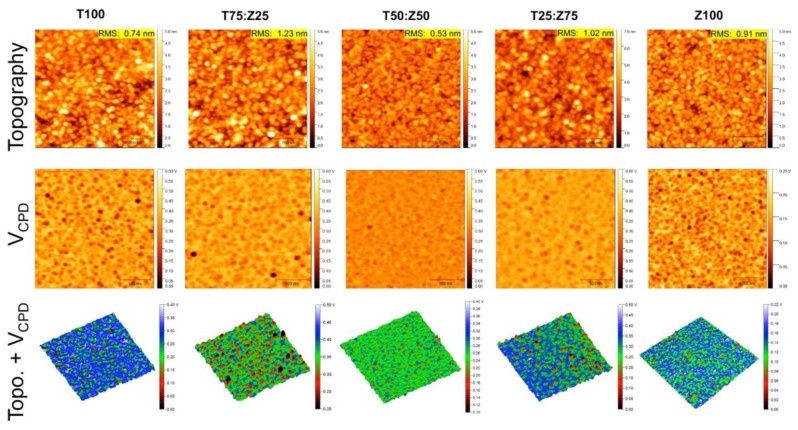
Topography, contact potential difference (V_CPD_) and 3D map highlight variation in local charge distribution of TiO_x_:Zn_1−x_Cd_x_S nanocomposite films. Scan area: 2μm×2μm.

**Figure 4 polymers-11-00980-f004:**
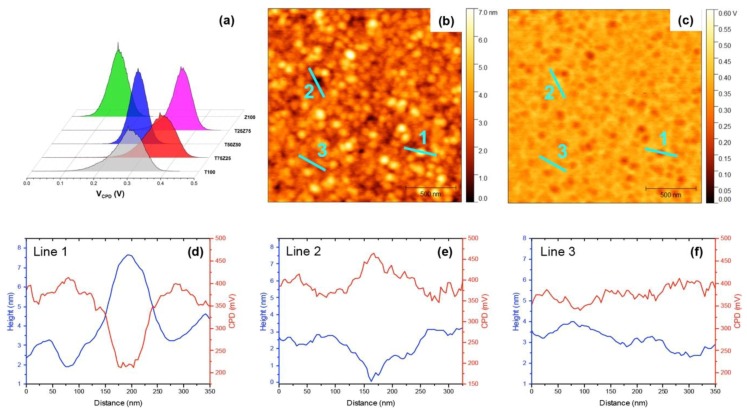
(**a**) Surface potential V_CPD_ histograms of TiO_x_:Zn_1−x_Cd_x_S nanocomposite films, (**b**) topography and (**c**) its corresponding surface potential image, (**d**–**f**) Profiles extracted from topography and surface potential at three different locations of T25:Z75 film.

**Table 1 polymers-11-00980-t001:** Performances of polymer solar cells fabricated from different T:Z ETL.

TiOx: ZnCdS Conc. (%)	*V_oc_* (V)	*J_sc_* (mA/cm^2^)	*FF*	*PCE* (%)
T100	0.879 (0.869 ± 0.008)	8.79 (8.67 ± 0.484)	0.55 (0.52 ± 0.028)	4.22 (3.94 ± 0.280)
T75:Z25	0.869 (0.867 ± 0.002)	10.59 (10.18 ± 0.465)	0.48 (0.47 ± 0.007)	4.40 (4.18 ± 0.245)
T50:Z50	0.910 (0.896 ± 0.012)	10.35 (10.25 ± 0.191)	0.53 (0.52 ± 0.013)	4.95 (4.77 ± 0.138)
T25:Z75	0.849 (0.844 ± 0.005)	8.55 (8.48 ± 0.214)	0.50 (0.50 ± 0.009)	3.63 (3.56 ± 0.086)
Z100	0.869 (0.863 ± 0.004)	8.69 (8.81 ± 0.166)	0.39 (0.38 ± 0.010)	2.93 (2.90 ± 0.030)

**Table 2 polymers-11-00980-t002:** Performances of perovskite solar cells fabricated from different T:Z ETL.

TiOx: ZnCdS Conc. (%)	*V_oc_* (V)	*J_sc_* (mA/cm^2^)	*FF*	*PCE* (%)
T100	0.859 (0.848 ± 0.007)	15.14 (15.72 ± 0.391)	0.60 (0.57 ± 0.015)	7.74 (7.65 ± 0.09)
T75:Z25	0.879 (0.880 ± 0.003)	17.31 (16.22 ± 1.922)	0.62 (0.62 ± 0.009)	9.47 (8.79 ± 1.024)
T50:Z50	0.889 (0.884 ± 0.003)	17.99 (17.82 ± 0.139)	0.61 (0.61 ± 0.001)	9.79 (9.67 ± 0.090)
T25:Z75	0.879 (0.875 ± 0.009)	15.19 (15.13 ± 1.177)	0.60 (0.57 ± 0.042)	8.07 (7.58 ± 0.590)
Z100	0.849 (0.844 ± 0.007)	15.19 (13.61 ± 3.159)	0.52 (0.50 ± 0.050)	6.74 (5.81 ± 1.796)

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
