# Peer review of "Low-Temperature Processed TiOx/Zn1−xCdxS Nanocomposite for Efficient MAPbIxCl1−x Perovskite and PCDTBT:PC70BM Polymer Solar Cells"

_polymers, 2019, doi:10.3390/polym11060980_

Round 1

Reviewer 1 Report

Most of the comments were addressed. Please check the spelling and thoroughly proof-read before publishing. 

Author Response

Response: We would like to thank the reviewer for giving us the opportunity to publish our manuscript. All of reviewer’s comments were addressed and added to the submitted manuscript. Some corrections have been made and added to this revised manuscript.

Reviewer 2 Report

The authors reported low temperature processed ETL for perovskite and polymer solar cells. The following issues should be addressed before it becomes publishable:

1. The thicknesses of each layer used in perovskite and polymer solar cells are important information that needs to be reported.

2. J-V curves of perovskite solar cells highly depends on the direction and the rate of the scan. Thus details about the measurement should be provided.

3. Perovskite solar cells have well-known hysteresis effects. Authors should study the J-V curve in detail by scanning in both directions and discuss if the new ETL and the interface have any effects on hysteresis.

4. J-V curves for polymer solar cells should be reported. A summary in Table 1 is not enough.

5. How many devices were tested for each type of solar cells? What is their distribution in terms of Voc, Jsc, FF and PCE?

6. Stability is an important aspect to evaluate solar cells. Authors should report device stability vs. time during operation under 1 sun illumination.

Author Response

Response to Reviewer 2 Comments

1. The thicknesses of each layer used in perovskite and polymer solar cells are important information that needs to be reported.

Response 1: We would like to thank the reviewer for giving us the opportunity to improve our manuscript. The thickness of each layer used in perovskite and polymer solar cells were added to the manuscript. For perovskite solar cells, the structure of ITO(300nm)/PEDOT:PSS(100nm)/ MAPbIxCl1-x (320nm)/PC70BM (80nm)/ETL(20nm)/Al(80nm) was fabricated. For polymer solar cell, the structure of ITO(300nm)/PEDOT:PSS(100nm)/PCDTBT:PC70BM(80nm) /ETL(20nm)/Al(80nm)was fabricated. We added the thickness of each layer in the manuscript as highlighted below.

“…20 μL of PEDOT:PSS (100nm) were coated twice by convective deposition at the speed of 3000 μm/s. The coated substrate was annealed on a hot plate at 120 °C for 30 min in the ambient environment (%RH 50-60). The polymer solar cell fabrication was continued under ambient condition while the PEDOT:PSS coated substrates were transferred to a glove box for perovskite solar cell fabrication.

For polymer solar cell with ITO/PEDOT:PSS/PCDTBT:PC70BM/TiOx/Al structure, 20 μL PCDTBT:PC70BM (80nm)  were convectively deposited at the speed of 750 μm/s and left in air for 3 min followed by the deposition of 20 μL TiOx solution with various Zn1-xCdxS suspension ratio. The coated films(20nm) were annealed at 80 oC.

For perovskite solar cell with ITO/PEDOT:PSS/CH3NH3PbIxCl3-x/PC70BM/TiOx/Al structure, 40 μL perovskite ink were deposited with spin coating technique with 2250 rpm for 30 s. The substrate and ink were kept at 70 °C before the deposition. The perovskite (300nm) coated substrate was annealed at 100 °C on a hot plate for 90 min. 35 μL of 50mg/mL PC70BM solution were spin casted onto the perovskite film at 1500 rpm for 30 s. 200 μL TiOx solution with various Zn1-xCdxS suspension ratio were spin casted (20nm) with 2500 rpm for 30 s. The multilayer films were then annealed at 80 °C for 30 min in a glove box.

Aluminum metal electrode (80 nm) was deposited on top of the TiOx layer to complete the polymer and perovskite solar cell fabrication…”

2. J-V curves of perovskite solar cells highly depends on the direction and the rate of the scan. Thus details about the measurement should be provided.

Response 2: We scanned the perovskite devices in the reverse direction from 1 V to -1V with 40 ms delay time and 10 mV step.  The details on the measurement were added to the manuscript as shown below.

“I-V measurements were conducted using source measurement unit (PXI-4130, National Instrument Inc with 10 mV scan step from 1V to -1V.”

and

“Negligible hysteresis for this p-i-n perovskite solar cell was observed after forward-reverse scan study as shown in Fig S2 in supplementary information. The results agree well with previous studies[40-42].”

Ref:

40          Levine, I.; Nayak, P.K.; Wang, J.T.-W.; Sakai, N.; Van Reenen, S.; Brenner, T.M.; Mukhopadhyay, S.; Snaith, H.J.; Hodes, G.; Cahen, D. Interface-Dependent Ion Migration/Accumulation Controls Hysteresis in MAPbI3 Solar Cells. J. Phys. Chem. C 2016, 120, 16399-16411, doi:10.1021/acs.jpcc.6b04233.

41.         Heo, J.H.; Han, H.J.; Kim, D.; Ahn, T.K.; Im, S.H. Hysteresis-less inverted CH3NH3PbI3 planar perovskite hybrid solar cells with 18.1% power conversion efficiency. Energy Environ. Sci. 2015, 8, 1602-1608, doi:10.1039/C5EE00120J.

42.         Habisreutinger, S.N.; Noel, N.K.; Snaith, H.J. Hysteresis Index: A Figure without Merit for Quantifying Hysteresis in Perovskite Solar Cells. ACS Energy Lett. 2018, 3, 2472-2476, doi:10.1021/acsenergylett.8b01627.

3. Perovskite solar cells have well-known hysteresis effects. Authors should study the J-V curve in detail by scanning in both directions and discuss if the new ETL and the interface have any effects on hysteresis.

Response 3: According to below literatures, hysteresis of p-i-n perovskite is much less than that of conventional n-i-p structure. We have tested the forward and reverse scan and found out that our ETL provide negligible hysteresis for both directions and different scan rates. The results of hysteresis study are shown in the supplemental information as Fig S2

Ref:

40            Levine, I.; Nayak, P.K.; Wang, J.T.-W.; Sakai, N.; Van Reenen, S.; Brenner, T.M.; Mukhopadhyay, S.; Snaith, H.J.; Hodes, G.; Cahen, D. Interface-Dependent Ion Migration/Accumulation Controls Hysteresis in MAPbI3 Solar Cells. J. Phys. Chem. C 2016, 120, 16399-16411, doi:10.1021/acs.jpcc.6b04233.

41.       Heo, J.H.; Han, H.J.; Kim, D.; Ahn, T.K.; Im, S.H. Hysteresis-less inverted CH3NH3PbI3 planar perovskite hybrid solar cells with 18.1% power conversion efficiency. Energy Environ. Sci. 2015, 8, 1602-1608, doi:10.1039/C5EE00120J.

42.       Habisreutinger, S.N.; Noel, N.K.; Snaith, H.J. Hysteresis Index: A Figure without Merit for Quantifying Hysteresis in Perovskite Solar Cells. ACS Energy Lett. 2018, 3, 2472-2476, doi:10.1021/acsenergylett.8b01627.

4. J-V curves for polymer solar cells should be reported. A summary in Table

1 is not enough.

Response 4: The J-V curves of polymer solar cells has been reported and added to the supporting information as Fig S1

5. How many devices were tested for each type of solar cells? What is their

distribution in terms of Voc, Jsc, FF and PCE?

Response 5: For each mixing ratio of TiOx:ZnCdS(T:Z), 4 devices were measured under 1 sun illumination. The distribution in terms of Voc, Jsc, FF and PCE were represented in terms of standard deviation as shown in Table 1 and 2.

6. Stability is an important aspect to evaluate solar cells. Authors should

report device stability vs. time during operation under 1 sun illumination.

Response 6:  We have added the device stability in supporting information as Fig S6 and Table S3. However, we observed the decay in efficiency for both pristine TiOx and ZnCdS mixed TiOx in ambient condition. The hygroscopic property of PEDOT:PSS as hole transporting layer for our solar cell causes the degradation of %PCE. The stability enhancement can be achieved by further adding cesium and bromide ions to perovskite structure. 

Round 2

Reviewer 2 Report

The authors have addressed the reviewer's comments properly and I recommend publication.

This manuscript is a resubmission of an earlier submission. The following is a list of the peer review reports and author responses from that submission.

Round 1

Reviewer 1 Report

This manuscript proposes low-temperature processed TiOx/Zn1-xCdxS nanocomposite for efficient MAPbIxCl1-x perovskite and PCDTBT:PC70BM polymer solar cells. The topic is interesting, and certainly consistent with the contents to be proposed to the readers of “Polymers”. However, the manuscript is not so well written and should be improved to be read with pleasure: this represents an important aspect in the current scenario of publications in international journals. Overall, I think that this manuscript could be accepted if the Authors will be able to take into account the following major revisions (in terms of bibliographic updates, grammar corrections and content deepening):

-          Detailed revisions: I spent several hours reading this manuscript, and Authors are asked to follow carefully the attached PDF file where I highlighted some points to be addressed. The attached file also contains language mistakes and typos (they are many in this work and should not be present when submitting a manuscript to an international journal: Authors are asked to check the manuscript better next time); some questions related to manuscript contents could also be present and Authors must consider them properly before submitting the revised manuscript. A point-by-point reply is required when the revised files are submitted.

-          Considering the amount of mistakes and typos present in this manuscript, a further check carried out by a native English speaker or by a professional English language center is suggested.

-          The Introduction should give a wider overview on the present scenario related to polymeric approaches for solar cells, both in terms of recently published reviews and research articles. In particular, applications in perovskite solar cells and dye solar cells are missing and a paragraph on this topic is highly suggested to be added in the Introduction. Authors are invited to go through the literature published in the last six months on these issues, and also on concepts developed some years ago in this field. Some of them are also mentioned in the above mentioned PDF file.

-          Authors should provide a clear explanation on the experimental error of the proposed research work. In particular, reproducibility of the phenomena described in the manuscript should be clearly stated in the “Results and Discussion” section; besides, some notes in the “Materials and Methods” section should be added highlighting which kind of experimental approach has been followed to check the reproducibility of the proposed system, the latter being of noteworthy importance in the present research field.

-          References: an article submitted to a journal should be consistent with the contents that it typically proposes in its table of contents. However, by checking the references of this manuscript, I did not find any articles published in this journal: this sounds rather strange. Maybe, Authors could check better the topics recently addressed by this journal, studying its table of contents and enriching the Introduction (as mentioned above) with some articles connected to this field.

Author Response

Response to Reviewer 1 Comments

Point 1: Detailed revisions: I spent several hours reading this manuscript, and Authors are asked to follow carefully the attached PDF file where I highlighted some points to be addressed. The attached file also contains language mistakes and typos (they are many in this work and should not be present when submitting a manuscript to an international journal: Authors are asked to check the manuscript better next time); some questions related to manuscript contents could also be present and Authors must consider them properly before submitting the revised manuscript. A point-by-point reply is required when the revised files are submitted.

Response 1: We would like to thank the reviewer for giving us the opportunity to improve our manuscript. We have carefully followed the suggestions and corrected our manuscript. Changes are highlighted in our revised manuscript.   

Point 2: Considering the amount of mistakes and typos present in this manuscript, a further check carried out by a native English speaker or by a professional English language center is suggested.

Response 2: This revised version has been checked by a professional English language service.

Point 3: The Introduction should give a wider overview on the present scenario related to polymeric approaches for solar cells, both in terms of recently published reviews and research articles. In particular, applications in perovskite solar cells and dye solar cells are missing and a paragraph on this topic is highly suggested to be added in the Introduction. Authors are invited to go through the literature published in the last six months on these issues, and also on concepts developed some years ago in this field. Some of them are also mentioned in the above-mentioned PDF file.

Response 3: We have added below paragraph in the introduction section to provide a boarder overview of polymeric approach in polymer, dye-sensitized and perovskite solar cell and included the reviewer’s recommended articles.

The ever-growing energy demand relies mainly on the combustion of fossil fuel, which continues to cause serious resource depletion and environmental pollution. Solar energy is a proven renewable energy that is environmentally friendly and free from regional restriction.  Inorganic solar cells (i.e. Si-based, gallium arsenide, copper indium gallium selenide, cadmium telluride, etc.) offer high efficiency (more than 25%) and stability, however, the technology development is hindered by sophisticated manufacturing process in addition to the handling of toxic materials [1]. While dye-sensitized, polymer and perovskite solar cells are the emerging photovoltaic devices that are both lightweight and low cost, they however have their own set of problems that inhibit their large-scale production as was highlighted in recent review papers [1-3]. To realize their practical applications, device stability is one of the critical factors that needs to be understood and overcome. Different approaches have been proposed for a more stable polymer [4-6], dye-sensitized and perovskite solar cells [7-11].

Point 4:  Authors should provide a clear explanation on the experimental error of the proposed research work. In particular, reproducibility of the phenomena described in the manuscript should be clearly stated in the “Results and Discussion” section; besides, some notes in the “Materials and Methods” section should be added highlighting which kind of experimental approach has been followed to check the reproducibility of the proposed system, the latter being of noteworthy importance in the present research field.

Response 4:  We have added references of experimental approach on material preparation in to the material and method section. In addition, statistical data of solar cell characterizations have been moved from the supporting information to the main text.

References

17.     Sun, Y.; Takacs, C.J.; Cowan, S.R.; Seo, J.H.; Gong, X.; Roy, A.; Heeger, A.J. Efficient, Air-Stable Bulk Heterojunction Polymer Solar Cells Using MoOx as the Anode Interfacial Layer. Advanced Materials 2011, 23, 2226-2230, doi:10.1002/adma.201100038

36.          Poormohammadi-Ahandani, Z.; Habibi-Yangjeh, A. Fast, green and template-free method for preparation of Zn1−xCdxS nanoparticles using microwave irradiation and their photocatalytic activities. Physica E: Low-dimensional Systems and Nanostructures 2010, 43, 216-223, doi:https://doi.org/10.1016/j.physe.2010.07.009.

37.           Gregory, D.G.; Lu, L.; Kiely, C.J.; Snyder, M.A. Interfacial Stabilization of Metastable TiO2 Films. J. Phys. Chem. C 2017, 121, 4434-4442, doi:10.1021/acs.jpcc.6b12943.

38.           Shi, Y.; Xing, Y.; Li, Y.; Dong, Q.; Wang, K.; Du, Y.; Bai, X.; Wang, S.; Chen, Z.; Ma, T. CH3NH3PbI3 and CH3NH3PbI3–xClx in Planar or Mesoporous Perovskite Solar Cells: Comprehensive Insight into the Dependence of Performance on Architecture. J. Phys. Chem. C 2015, 119, 15868-15873, doi:10.1021/acs.jpcc.5b02784.

39.          Sanglee, K.; Chuangchote, S.; Chaiwiwatworakul, P.; Kumnorkaew, P. PEDOT:PSS Nanofilms Fabricated by a Nonconventional Coating Method for Uses as Transparent Conducting Electrodes in Flexible Electrochromic Devices. Journal of Nanomaterials 2017, 2017, 8, doi:10.1155/2017/5176481

Point 5: References: an article submitted to a journal should be consistent with the contents that it typically proposes in its table of contents. However, by checking the references of this manuscript, I did not find any articles published in this journal: this sounds rather strange. Maybe, Authors could check better the topics recently addressed by this journal, studying its table of contents and enriching the Introduction (as mentioned above) with some articles connected to this field.

Response 5:   We have included below references that are related to our work and published by the Polymers journal in the literature review.

References

1.             Hou, W.; Xiao, Y.; Han, G.; Lin, J.-Y. The Applications of Polymers in Solar Cells: A Review. Polymers 2019, 11, 143.

2.             Biniek, L.; Nielsen, C.B. Organic Photovoltaics: More than Ever, an Interdisciplinary Field. Polymers 2016, 8, 70.

20.            Jiang, M.; Niu, Q.; Tang, X.; Zhang, H.; Xu, H.; Huang, W.; Yao, J.; Yan, B.; Xia, R. Improving the Performances of Perovskite Solar Cells via Modification of Electron Transport Layer. Polymers 2019, 11, 147, doi:10.3390/polym11010147.

Reviewer 2 Report

1)    English should be taken care of.

2)    Abstract : Need some refining and polishing.

3)    Authors should identify many high quality references and include in appropriate places.

4)    Introduction: I suggest the author to cite relevant articles focusing on the buffer/interfacial layer (such as Energy Environ. Sci., 2011,4, 285-310, Solar Energy Materials and Solar Cells 174, 112-123, Solar Energy Materials and Solar Cells 153, 148-163, Solar Energy Materials and Solar Cells 141, 275-290, Scientific reports 7, 45079 etc.)

5)    65 to 67 sentences need attention.

6)    Materials and Methods:

a.     2.1.3 and 1.4, need to change title (like photoactive ink formulation)

7)    I suggest the authors to move the device statistics for polymer/perovskite solar cells into main text. If possible, also consider moving HRTEM and SAED patterns.

8)    Equations are not clear

9)    Please enlarge Fig 1 (c), (d) and (e), Fig. 2 (a) (b) as it’s not clear           

10)  Please remove unwanted spaces in the manuscript.

11)  Take care of fonts, formatting and indentation.

Author Response

Response to Reviewer 2 Comments

Point 1:  English should be taken care of.

Response 1:  This revised version has been checked by a professional English language service.

Point 2: Abstract: Need some refining and polishing.

Response 2:  The abstract was modified as shown below.

Majority of high-performance perovskite and polymer solar cells consist of a TiO2 electron transport layer (ETL) processed at high temperature (>450 oC). Here, we demonstrate that low-temperature (80 oC) ETL thin film of TiOx:Zn1-xCdxS can be used as an effective ETL and its band energy can be tuned by varying the TiOx:Zn1-xCdxS ratio. At the optimal ratio of 50:50 (vol%), the MAPbIxCl1-x perovskite and PCBTBT:PC70BM polymer solar cells achieved 9.79% and 4.95%, respectively. Morphological and optoelectronic analyses show that tailoring band edges and homogeneous distribution of the local surface charges can improve the solar cells efficiency by more than 2%. We proposed a plausible mechanism to rationalize the variation in morphology and band energy of the ETL.   

Point 3: Authors should identify many high-quality references and include in appropriate places.

Response 3:    We have added the below references in our manuscript.

References

1.             Hou, W.; Xiao, Y.; Han, G.; Lin, J.-Y. The Applications of Polymers in Solar Cells: A Review. Polymers 2019, 11, 143.

2.             Biniek, L.; Nielsen, C.B. Organic Photovoltaics: More than Ever, an Interdisciplinary Field. Polymers 2016, 8, 70.

3.             Malinkiewicz, O.; Yella, A.; Lee, Y.H.; Espallargas, G.M.; Graetzel, M.; Nazeeruddin, M.K.; Bolink, H.J. Perovskite solar cells employing organic charge-transport layers. Nature Photonics 2014, 8, 128-132, doi:10.1038/nphoton.2013.341.

4.             Sai-Anand, G.; Gopalan, A.-I.; Lee, K.-P.; Venkatesan, S.; Qiao, Q.; Kang, B.-H.; Lee, S.-W.; Lee, J.-S.; Kang, S.-W. Electrostatic nanoassembly of contact interfacial layer for enhanced photovoltaic performance in polymer solar cells. Solar Energy Materials and Solar Cells 2016, 153, 148-163, doi:10.1016/j.solmat.2016.04.018.

5.             Sai-Anand, G.; Gopalan, A.-I.; Lee, K.-P.; Venkatesan, S.; Kang, B.-H.; Lee, S.-W.; Lee, J.-S.; Qiao, Q.; Kwon, D.-H.; Kang, S.-W. A futuristic strategy to influence the solar cell performance using fixed and mobile dopants incorporated sulfonated polyaniline based buffer layer. Solar Energy Materials and Solar Cells 2015, 141, 275-290, doi:10.1016/j.solmat.2015.05.035.

6.             Gopalan, S.-A.; Gopalan, A.-I.; Vinu, A.; Lee, K.-P.; Kang, S.-W. A new optical-electrical integrated buffer layer design based on gold nanoparticles tethered thiol containing sulfonated polyaniline towards enhancement of solar cell performance. Solar Energy Materials and Solar Cells 2018, 174, 112-123, doi:10.1016/j.solmat.2017.08.029.

7.             Bella, F.; Lamberti, A.; Bianco, S.; Tresso, E.; Gerbaldi, C.; Pirri, C.F. Floating Photovoltaics: Floating, Flexible Polymeric Dye-Sensitized Solar-Cell Architecture: The Way of Near-Future Photovoltaics (Adv. Mater. Technol. 2/2016). Advanced Materials Technologies 2016, 1, doi:10.1002/admt.201670011.

8.             Wang, D.; Zhang, L.; Deng, K.; Zhang, W.; Song, J.; Wu, J.; Lan, Z. Influence of Polymer Additives on the Efficiency and Stability of Ambient-Air Solution-Processed Planar Perovskite Solar Cells. Energy Technology 2018, 6, 2380-2386, doi:10.1002/ente.201800378.

9.             Abate, A.; Correa‐Baena, J.-P.; Saliba, M.; Su'ait, M.S.; Bella, F. Perovskite Solar Cells: From the Laboratory to the Assembly Line. Chemistry – A European Journal 2018, 24, 3083-3100, doi:10.1002/chem.201704507.

10.           Imperiyka, M.; Ahmad, A.; Hanifah, S.A.; Bella, F. A UV-prepared linear polymer electrolyte membrane for dye-sensitized solar cells. Physica B: Condensed Matter 2014, 450, 151-154, doi:10.1016/j.physb.2014.05.053.

11.           Han, T.-H.; Lee, J.-W.; Choi, C.; Tan, S.; Lee, C.; Zhao, Y.; Dai, Z.; De Marco, N.; Lee, S.-J.; Bae, S.-H., et al. Perovskite-polymer composite cross-linker approach for highly-stable and efficient perovskite solar cells. Nature Communications 2019, 10, 520, doi:10.1038/s41467-019-08455-z.

12.           Bai, Y.; Meng, X.; Yang, S. Interface Engineering for Highly Efficient and Stable Planar p‐i‐n Perovskite Solar Cells. Advanced Energy Materials 2018, 8, doi:10.1002/aenm.201701883.

Point 4:  Introduction: I suggest the author to cite relevant articles focusing on the buffer/interfacial layer (such as Energy Environ. Sci., 2011,4,

285-310, Solar Energy Materials and Solar Cells 174, 112-123, Solar Energy Materials and Solar Cells 153, 148-163, Solar Energy

Materials and Solar Cells 141, 275-290, Scientific reports 7, 45079 etc.)

Response 4:  We have included the reviewer recommended references in the introduction.

Point 5: 65 to 67 sentences need attention.

Response 5:  We have modified this sentence as follows.

Original sentence

“Work function of ETL composite films obtained from SKPM can be used in complement with UPS results and photovoltaic performance to shed more light into the fundamental understanding of electron transport layer in solar cells”

Modified sentence

            “The local morphology with corresponding work function obtained from SKPM can shed more light into the fundamental understanding of electron transport layer in solar cells.”

Point 6: Materials and Methods:

a. 2.1.3 and 1.4, need to change title (like photoactive ink formulation)

Response 6:  We have changed the title of 2.1.3 and 2.1.4 to polymer ink formulation and perovskite ink formulation respectively.      

Point 7: I suggest the authors to move the device statistics for polymer/perovskite solar cells into main text. If possible, also consider moving HRTEM and SAED patterns.

Response 7:  We have moved the device statistics for both polymer/perovskite solar cells to the main text. Here, we like to highlight the fact that for the first time, data collected from nanoscale (SKPM technique) and macroscale (UPS technique) work function are implementing each other and providing insights into the device’s performance. We prefer to keep the HRTEM and SAED patterns in supporting information to make our main data more concise in the main text.

Point 8: Equations are not clear

Response 8:  All equations have been modified for better clarity

Point 9: Please enlarge Fig 1 (c), (d) and (e), Fig. 2 (a) (b) as it’s not clear

Response 9:  Fig 1 and Fig 2. have been modified for better resolution

Point 10: Please remove unwanted spaces in the manuscript.

Response 10:  We have removed the unwanted spaces in the manuscript.

Point 11: Take care of fonts, formatting and indentation.

Response 11:  The fonts, formatting and indentation throughout the manuscript have been adjusted.

Reviewer 3 Report

In this work, the authors demonstrated the application of a low-temperature (80 °C) processed ETL in perovskite and organic solar cells. As a result, the best PCEs of perovskite (MA-based) and organic solar cells (PCBTBT:PC70BM) based on the new ETL were 9.79% and 4.95%, respectively. In my opinion, the novelty and importance of this work can not reach the publishing standards. Please reject this work.

1. The state-of-the-art PCEs of MA-based perovskite and PCBTBT/PC71BM solar cells can reach to 20% and 7%, respectively, which are much higher than the result presented by the authors.

2. The authors claimed low-temperature (80 °C) processed ETL, however, the processed temperature of commercialized ZnO nanoparticles ETL is only 60 °C.

3. The mechanism of energy level tuning should be addressed.

4. As a device paper, a table of photovoltaic properties should be added.

Author Response

Response to Reviewer 3 Comments

Point 1: The state-of-the-art PCEs of MA-based perovskite and PCBTBT/PC71BM solar cells can reach to 20% and 7%, respectively, which are much higher than the result presented by the authors.

Response 1:  We are fully aware that MA-based Perovskite and PCBTBT based solar cell achieved 20% and 7%, respectively. While significant effort has been paid in achieving high efficiency, technology viability also depends on device stability, process feasibility and fundamental insights. Here, we focused on developing ETL material for p-i-n solar cell device with PEDOT:PSS as hole transporting layer. The hygroscopic property of PEDOT:PSS cause the lower stability and lower efficiency. In addition, our polymer solar cells were fabricated out of glove box with humidity above 50% RH and still achieved 5% PCE. We also provided the further insights into distribution of local surface charges and established correlation between local electronic properties, morphology of the nanocomposite ETL films and performance of devices, which have not been reported before. 

Point 2: The authors claimed low-temperature (80 °C) processed ETL, however, the processed temperature of commercialized ZnO nanoparticles ETL is only 60 °C.

Response 2:  The reviewer raised an interesting point. Although the temperature of commercialized ZnO is only 60 °C, the processed temperature of TiOx/TiO2 is well documented above that temperature. We have shown in our abstract and introduction a brief overview on current processed temperature for TiOx/TiO2. The performance of ZnO and TiO2 as an ETL is still under debate and is not in the scope of our work. Here, our effort here is to bring down the processed temperature for titania to make the technology more feasible. Our TiOx were prepared at room temperature and heated up to 80 oC during the fabrication. ZnO nanoparticle sometime needs hydrothermal synthesis at temperature more than 90 oC and then dried above 100oC before using in solar cell fabrication. Here, we used 80oC in thermal annealing of our TiOx to make sure that solvent was evaporated.

Point 3: The mechanism of energy level tuning should be addressed.

Response 3:  Mechanism of energy level tuning was discussed in result and discussion section as follows.

We suggest the following mechanism to rationalize for morphological and electronic variation of TiOx:Zn1-xCdxS nanocomposite films. In pure TiOx film, nanocrystals can be formed from Ti-O monomers via hydrolysis and condensation reactions which are strongly dependent on reaction temperature. Solvent evaporates, supersaturates the polymeric precursor solution and leads to the formation of a large number of Ti-O species. At low temperature (80°C), the nuclei impinge on each other and impede further grain growth via diffusion, resulting in a relatively smooth film of small grains (RMS: 0.7 nm), but the mixture of amorphous and crystalline phases (dominantly amorphous phase) leads to broad distribution of work function (3.47 ± 0.44 eV). Pure Zn1-xCdxS film follows similar mechanism as TiOx film, however, there is no polymeric precursor. The pure Zn1-xCdxS film is in smooth, crystalline film (RMS: 0.9 nm), resulting in narrow distribution of work function (3.77 ± 0.29V). When preexisting Zn1-xCdxS seeds are present in the initial stage of the synthesis, the Ti-O species are immobilized onto the seed crystals to overcome the activation energy barrier for nucleation, leading to the formation of bigger grains compared to that of pure films and creating an ordered electronic network of TiOx surrounding the grains. For T75Z25 film, the density of Zn1-xCdxS in the composite film is too low, resulting in rougher film due to scattering of the large crystalline grains. For T25Z75 film, the density of Zn1-xCdxS in the composite film is too high, the grains are interconnected by the polymer chains similar to the mechanism proposed by Yang’s group [11], however, the volume of TiOx is not enough to bridge the gaps between grains, resulting in a rough surface. For T50Z50 film, the density of Zn1-xCdxS is optimal in the TiOx matrix, the distribution of seed crystals and its proximity interaction with surrounding polymer chains is the most effective, resulting in the smoothest film among all. As the morphology and internal structures of ETL films are tuned by addition of Zn1-xCdxS, the band energy of ETL is simultaneously modified due to presence of Zn1-xCdxS energy states and surrounding TiOx electronic network, which are supported by UV-Vis, UPS and SKPM results. Ultimately, appropriate addition of Zn1-xCdxS can be used to maximize the optoelectronic output and stability of ETL film and the corresponding devices.

Point 4: As a device paper, a table of photovoltaic properties should be added.

Response 4:  We have moved the device statistics for both polymer/perovskite solar cells from the supporting information to the main text.

Reviewer 4 Report

The authors have reported perovskite and polymer solar cells with a low-temperature processed ETL layer. This work is interesting and could be publishable after addressing the following issues.

Device stability is an important aspect for solar cells. Authors should study the stability of the fabricated solar cells, and show how efficiency, Fill Factor, Jsc and Voc changes versus time under constant sun illumination.

Authors should conduct a statistic study on the device performance and report efficiency variations from sample to sample.

Efficiencies change by varying TiOx and ZnCdS ratio with an optimized ratio in 50:50. Authors should provide more discussions on why the other ratios will decrease the performance. The authors have conducted several experiments, while more insights regarding the mechanisms are required.

Author Response

Response to Reviewer 4 Comments

Point 1: Device stability is an important aspect for solar cells. Authors should study the stability of the fabricated solar cells, and show how efficiency, Fill Factor, Jsc and Voc changes versus time under constant sun illumination.

Response 1:  We have added the device stability in supporting information. However, we observed the decay in efficiency for both pristine TiOx and ZnCdS mixed TiOx. The hygroscopic property of PEDOT:PSS as hole transporting layer for our solar cell causes the degradation of %PCE.

Point 2:Authors should conduct a statistic study on the device performance and report efficiency variations from sample to sample. Efficiencies change by varying TiOx and ZnCdS ratio with an optimized ratio in 50:50.

Response 2:   We did conduct a statistic study on the device performance and efficiency variations for all samples. The data were listed in the supporting information. For clarification, we have moved the device statistics for both polymer/perovskite solar cells from the supporting information to the main text.

Point 3:Authors should provide more discussions on why the other ratios will decrease the performance. The authors have conducted several experiments, while more insights regarding the mechanisms are required

Response 3:  We have added the discussion of the effect of ZnCdS ratio on the device performance in the discussion section as follows.

We suggest the following mechanism to rationalize for morphological and electronic variation of TiOx:Zn1-xCdxS nanocomposite films. In pure TiOx film, nanocrystals can be formed from Ti-O monomers via hydrolysis and condensation reactions which are strongly dependent on reaction temperature. Solvent evaporates, supersaturates the polymeric precursor solution and leads to the formation of a large number of Ti-O species. At low temperature (80°C), the nuclei impinge on each other and impede further grain growth via diffusion, resulting in a relatively smooth film of small grains (RMS: 0.7 nm), but the mixture of amorphous and crystalline phases (dominantly amorphous phase) leads to broad distribution of work function (3.47 ± 0.44 eV). Pure Zn1-xCdxS film follows similar mechanism as TiOx film, however, there is no polymeric precursor. The pure Zn1-xCdxS film is in smooth, crystalline film (RMS: 0.9 nm), resulting in narrow distribution of work function (3.77 ± 0.29V). When preexisting Zn1-xCdxS seeds are present in the initial stage of the synthesis, the Ti-O species are immobilized onto the seed crystals to overcome the activation energy barrier for nucleation, leading to the formation of bigger grains compared to that of pure films and creating an ordered electronic network of TiOx surrounding the grains. For T75Z25 film, the density of Zn1-xCdxS in the composite film is too low, resulting in rougher film due to scattering of the large crystalline grains. For T25Z75 film, the density of Zn1-xCdxS in the composite film is too high, the grains are interconnected by the polymer chains similar to the mechanism proposed by Yang’s group [11], however, the volume of TiOx is not enough to bridge the gaps between grains, resulting in a rough surface. For T50Z50 film, the density of Zn1-xCdxS is optimal in the TiOx matrix, the distribution of seed crystals and its proximity interaction with surrounding polymer chains is the most effective, resulting in the smoothest film among all. As the morphology and internal structures of ETL films are tuned by addition of Zn1-xCdxS, the band energy of ETL is simultaneously modified due to presence of Zn1-xCdxS energy states and surrounding TiOx electronic network, which are supported by UV-Vis, UPS and SKPM results. Ultimately, appropriate addition of Zn1-xCdxS can be used to maximize the optoelectronic output and stability of ETL film and the corresponding devices.

Round 2

Reviewer 1 Report

The manuscript has been properly revised and I recommend its publication.

Reviewer 2 Report

I appreciate authors considerable efforts in improving the quality of the manuscript. Most of my comments were addressed.

Reviewer 3 Report

In the response letter, the authors can't address my concerns (point 1 and 2). The performance of solar cells are very sensitive, if the authors want to claim something, they MUST achieve the performance baseline of device first. Otherwise, we can not obtain any information from their work. For example, if the PCE baseline of a OPV is 7%, the author increase the PCE from 3% to 5% by a new method. We can not claim the new method is "working". If the author can increase the PCE from 7% to 8%, it's working.

Reviewer 4 Report

The authors have addressed all the concerns raised by the reviewer and this work is publishable now.